

## M2 Monthly and annual mode 1 and mode 2 internal tide atlases from altimetry data and MIOST: focus on the Indo-Philippine Archipelago and the region off the Amazon shelf.

Michel Tchilibou[1], Simon Barbot[2], Loren Carrere[1], Ariane Koch-Larrouy[2], Gérald Dibarboure[3] and Clément Ubelmann[4]

[1] Collecte Localisation Satellites, 31520 Ramonville-Saint-Agne, France

[2] Université de Toulouse, LEGOS (CNES/CNRS/IRD/UT3), 31400 Toulouse, France

[3] Centre National d'Etude Spatiales, 31400, Toulouse, France

[4] Datlas, Grenoble, France

Correspondence to: Michel Tchilibou (mtchilibou@groupcls.com)

**Abstract**

The M2 MIOST24 (Multivariate Inversion of Ocean Surface Topography 2024) internal tide atlases are available in annual (MIOST24a) and monthly (MIOST24m) regional versions for modes 1 and 2 in the Indo-Philippine archipelago and the region off the Amazon shelf. They are derived from 25 years (1993-2017, period 1) of sea level anomalies (SLA) from altimetry observations. Compared to MIOST22, MIOST24 incorporates M2 modes 1 and 2 wavelengths based on monthly stratification profiles from GLORYS12v1 (1993-2020). The differences between MIOST24a and MIOST22a lead to RMSE (Root Mean Square Error) of up to 3 cm, reflecting amplitude changes, while the RMSE between MIOST24a and HRET (High-Resolution Empirical Tide) indicate both amplitude and phase differences. MIOST24m highlights significant monthly variability of M2 internal tides in the Indo-Philippine archipelago and off the Amazon shelf. In the Amazon region, the internal tide propagates far offshore from March to June but is blocked closer to the coast from August to December. For both regions, mode 1 monthly variability is mainly phase-related, while mode 2 is more amplitude-dependent. Variance reductions of SLA show that MIOST24m outperforms MIOST24a, MIOST22a and HRET on period 1 in the two regions of interest, and on the Amazon region in period 2 (2018-2023). Monthly atlases are therefore recommended to correct the internal tides of the SLA used to derive them. All these results support the development of an improved MIOST24 global atlas.

**Introduction**

Since the early 1990s and the launch of the TOPEX/Poseidon mission, spatial altimetry has provided an independent source of measurements of ocean surface topography. These measurements are essential for a better understanding of ocean surface dynamics, deep ocean circulation and their impact on climate (Escudier et al., 2017). Significant progress has been made in the processing and reprocessing of altimetry data (Dibarboure et al. 2011, Ablain et al., 2015, Pujol et al., 2016, Pujol et al.,2023). Today, more than 30 years of observations are available. They can be used directly or indirectly (model constraints, data assimilation) to generate global atlases of barotropic tides (Stammer et al., 2014, Carrere et al., 2016, Lyard et al., 2020, Lyard et al.,2024 Desai and Ray, 2014, Egbert and Erofeeva, 2002) and baroclinic tides, also known as internal tides (Dushaw et al., 2011; Dushaw, 2015, Zhao et al.,2012, Ray and Zaron, 2016; Zaron et al., 2019, Ubelmann et al., 2022). In turn, these tidal atlases are used as geophysical corrections in the Data Unification and Altimeter Combination System (DUACS) chain to compute the Sea Level Anomaly (SLA) along the altimeter tracks (L3 product) and on regular horizontal grids (L4 product).



Internal tide (IT) atlases have been included in the geophysical corrections of the altimetry dataset since
the transition from version DT2018 to DT2021 in the DUACS processing chain (Sanchez-Roman et al., 2021;
Lievin et al., 2020; Faugere et al., 2022). The internal tide model currently used as a reference in DUACS is
HRETv8.1 (High-Resolution Empirical Tide 8.1, HRET in the rest of the paper) by Zaron (2019). Several global
atlases exist as described in (Carrère et al., 2021), such as those by Zhao (Zhao et al., 2012, 2018, 2019, 2021)
or MIOST (Multivariate Inversion of Ocean Surface Topography; Ubelmann et al., 2021, 2022). The MIOST
atlas of Ubelmann et al. (2022), like the Zhao atlases, uses mode 1 and 2 theoretical internal tide wavelengths'
climatologies to extract the internal tide signal from the altimetric SLA. For the 2022 version of the MIOST
atlas, Ubelmann et al. (2022) use the first Rossby deformation radius climatology from Chelton et al. (1998)
as an approximation of the mode 1 wavelength and divide it by two to approximate the mode 2 wavelength.
These approximations are not always accurate because each baroclinic mode has characteristics influenced
by ocean stratification (Gerkerma et al., 2004). Moreover, the climatology of Chelton et al. (1998) is based
on a very different period from the 1993 - 2017 period of the altimetry data used by Ubelmann et al. (2022),
which could lead to inaccuracies in the location and amplitude of internal tides. The first objective of this
study is to propose a 2024 version of MIOST IT atlas based on more appropriate mode 1 and 2 wavelengths,
thus overcoming the deficiencies of the wavelengths prescribed in Ubelmann et al. (2022).
The second objective of this study concerns the monthly variability of the internal tide and the relevance
of replacing the HRET internal tide atlas with monthly atlases in the DUACS chain. Altimetry-derived internal
tide atlases only include the stationary (or coherent, or phase-lock) part of the internal tides. The coherent
and incoherent (non-stationary) internal tides both vary with stratification and the interactions of the internal
tides with the ocean circulation, including mesoscale eddies (Tchilibou et al., 2020; Duda et al.,2018; Dunphy
et al., 2014). This variability is not considered in annual atlases such as HRET, which use the altimetry
database as a single set. Zhao (2021) builds subsets of altimetry data over the four meteorological seasons
and shows that seasonal atlases perform better in tropical regions but with some limitations. In this study,
we extend the question of internal tide variability to the monthly scale, focusing on the M2 wave in the Indo-
Philippine archipelago (Figure 1a) and the region of the Amazon shelf in the tropical Atlantic (Figure 1b).

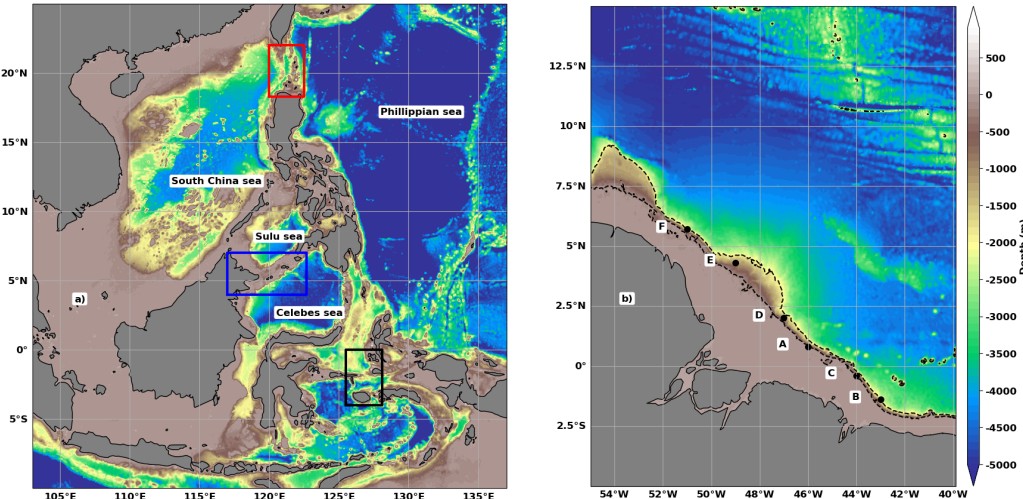


**Figure 1:** Bathymetric depth (in m) in the Indo-Philippine archipelago (a) and the region off the Amazon shelf
(b). The major internal tide generation areas are indicated by red (Luzon Strait), blue (Sulu Island Chain) and
black (Seram Sea) boxes for the Indo-Philippine archipelago and by the letters A-F for the Amazon shelf-
break.



These two tropical regions are chosen because they are hotspots of M2 internal tide generation and the
site of high M2 internal tide variability (Jan et al.,2008; Muller et al.,2012; Ray and Zaron 2011, Niwa and
Hibiya 2001, 2004 and 2014, Arbic et al.,2012; Nugroho et al., 2018; Zhao and Qui 2023; Pickering et al.,2015;
Rainville et al.,2013; Tchilibou et al., 2022; Assene et al., 2024). In addition, the Indo-Philippine archipelago
is a transit zone between the Pacific and Indian Oceans (Sprintall et al., 2014; Hurlburt et al., 2011), and the
tide-induced mixing there affects the coupled ocean-atmosphere system and thus the global climate system
(Koch-Larrouy et al., 2010). More details on these two regions can be found in Nugroho (2018) and Tchilibou
et al. (2022). The paper is divided into 5 sections. In section 1 we give an overview of HRET, compare the
wavelengths' bases of the 2022 and 2024 versions of MIOST-IT, and present the altimetry data and how they
are organized to derive the annual (classical) and monthly atlases. In section 2, the annual atlases of MIOST-
IT 2022 and 2024 are compared qualitatively with HRET. The efficiency of the monthly atlases in correcting
for internal tides in the altimetry data is analyzed in section 4, and the paper is concluded in section 5.
**1-   Internal tide atlases and data:**
In this section, we describe the different internal tide atlases developed within the study (MIOST24) as
well as the datasets used for computation, and the other atlases used for comparison (MIOST22 and
HRET).
**1.1- HRET**
HRET (for HRETv8.1; Zaron 2019) is an empirical atlas of internal tides at the M2, S2, K1 and O1 frequencies,
developed from the analysis of the 1993 to 2017 (25 years) exact repeat mission altimetry data (Topex, Jason
1 to 3, ERS, Envisat, Saral AltiKa and the GEOSAT follow-on). The method used to construct the HRET internal
tide atlas involves a local two-dimensional Fourier analysis of the along-track data, and a least-squares fit by
a second-order polynomial. HRET is provided on a horizontal grid of 0.05°X0.05° (1/20°) and includes modes
1 and 2. A mask is applied in the regions where the amplitude of the internal tides is very noisy.
**1.2- MIOST22 and MIOST24 mode 1 and 2 wavelengths for M2:**
MIOST is an empirical atlas of internal tides obtained by a single inversion that simultaneously separates
mesoscale and internal tides (modes 1 and 2) from altimetry observations (Ubelmann et al., 2022). The
2022 and 2024 versions of MIOST-IT atlases are hereafter referred to as MIOST22 and MIOST24. The
suffixes 'a' and 'm' are used to distinguish between annual (e.g. MIOST24a) and monthly (e.g. MIOST24m)
MIOST atlases.
In MIOST, mesoscales are expressed as a reduced wavelet basis. Internal tides are defined by a plane-
wave basis according to the dispersion relation (e.g. Rainville et Pinkel, 2006) given in Equation 1, where $f$
is the Coriolis parameter, $\lambda_n$, $C_n$ and $\omega$ are the wavelength, eigenspeed and pulsation of the internal tides
of mode $n$, respectively.

$$\lambda_n = \frac{2\pi C_n}{\sqrt{\omega^2 - f^2}}, (1)$$

As mentioned in the introduction, for MIOST22 the wavelength of mode 1 corresponds to the first Rossby
deformation radius and the wavelength of mode 2 is half the wavelength of mode 1. For MIOST24, mode 1
and 2 wavelengths are determined independently by solving the eigenvalue equation (Eq 2; Gill, 1982) with
the boundary conditions $\phi_n(0) = \phi_n(H) = 0$, where H is the ocean depth and $\phi_n(z)$ the modal vertical
structure.



$$\frac{\partial^2 \phi_n(z)}{\partial z^2} + \frac{N(z)^2}{C_n^2} \phi_n = 0, (2)$$

The $N(z)$ stratification profiles required for equation 2 are computed from the 1993 - 2020 monthly
climatologies of the potential temperature and salinity fields from the GLORYS12v1 reanalysis
(https://doi.org/10.48670/moi-00021, last accessed 30/07/2024). GLORYS12v1 is a global CMEMS
(Copernicus Marine Environment Monitoring Service) product with a horizontal resolution of 1/12° and 50
levels. It is based on an eddy-resolving NEMO platform assimilating along tracks altimetric SLA, satellite sea
surface temperature observations and in situ vertical temperature and salinity profiles (Argo, moorings, etc.).
The GLORYS12v1 reanalyses are used as they are to calculate $N(z)$ and derive the monthly mode 1 and 2
wavelengths of the M2 internal tides required for MIOST24m. The MIOST24a atlas requires annual
wavelengths, obtained using $N(z)$ derived from the annual mean of GLORYS12v1 fields. The annual
wavelengths used for MIOST22a and MIOST24a and the standard deviations of the monthly wavelengths
used for MIOST24m are shown in Figure 2 for the Indo-Philippine Archipelago and Figure 3 for the region off
the Amazon shelf.
In the Indo-Philippine archipelago, the annual mode 1 (Figures 2a and 2c) and mode 2 (Figures 2b and 2d)
M2 wavelengths used for MIOST22 (Figures 2a and 2b) are generally larger than those used for MIOST24
(Figure 2c and 2d). The differences can be up to 10 km and are quite pronounced south of the Philippine Sea,
in the South China Sea, in the Celebes Sea and the Banda Sea. Only in the Sulu Sea do MIOST24 wavelengths
exceed those of MIOST22. The simplified approximation that the wavelength of mode 2 is half the one of
mode 1, used for MIOST22, implies that the spatial distributions of the two modes are equivalent (Figure 2a
and 2b).  We assume that the locations of the maximum wavelengths in mode 1 also correspond to the
locations of the longest wavelengths in mode 2, and similarly for the locations of the shortest wavelengths.
This is not the case when looking at the MIOST24 wavelength maps for modes 1 (Figure 2c) and 2 (Figure 2d).
In the Sulu Sea, the mode 2 wavelengths of MIOST22 are about 50 km, *i.e.* among the shortest, while in
MIOST24 they are of the order of 80 km, *i.e.* the highest.

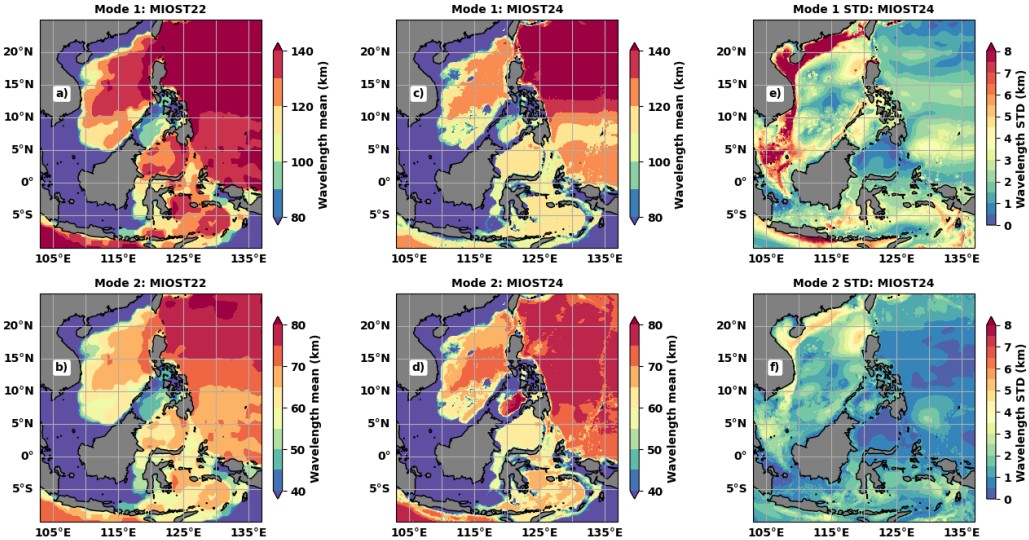


**Figure 2:** M2 wavelengths values (a-d, in km) and standards deviations (e-f, in km) in the Indo-Philippine
archipelago: (a) mode 1 and (b) mode 2 annual values used for MIOST22a,  (c) mode 1 and (d) mode 2 annual
values used for MIOST24a, standard deviation of  (e) mode 1 and  (f) mode 2 monthly wavelengths used for
MIOST24m.



Off the Amazon shelf, the M2 wavelengths of MIOST22 and MIOST24 show differences in both spatial
distribution and values (Figure 3). In MIOST22, mode 1 (Figure 3a), and mode 2 (Figure 3b), the wavelength
gradient is positive from offshore to about 7°N. Thereafter the wavelengths decrease slightly. In MIOST24
the structures are more homogeneous (Figure 3c and 3d). Outside the continental shelf, mode 1 wavelengths
(Figure 3c) are mostly between 100 and 110 km, while they exceed 120 km in MIOST22 (Figure 3a). For mode
2 (Figure 3d), the dominant structure corresponds to wavelengths between 65 and 70 km, while in MIOST22
the space bounded by this structure also includes a short-wavelength zone between 60 and 65 km (Figure
3c). As in the Indo-Philippine archipelago, there is no direct relationship between the spatial distributions of
mode 1 and 2 annual wavelengths in MIOST24. This confirms that the wavelength characteristics are specific
to each mode.

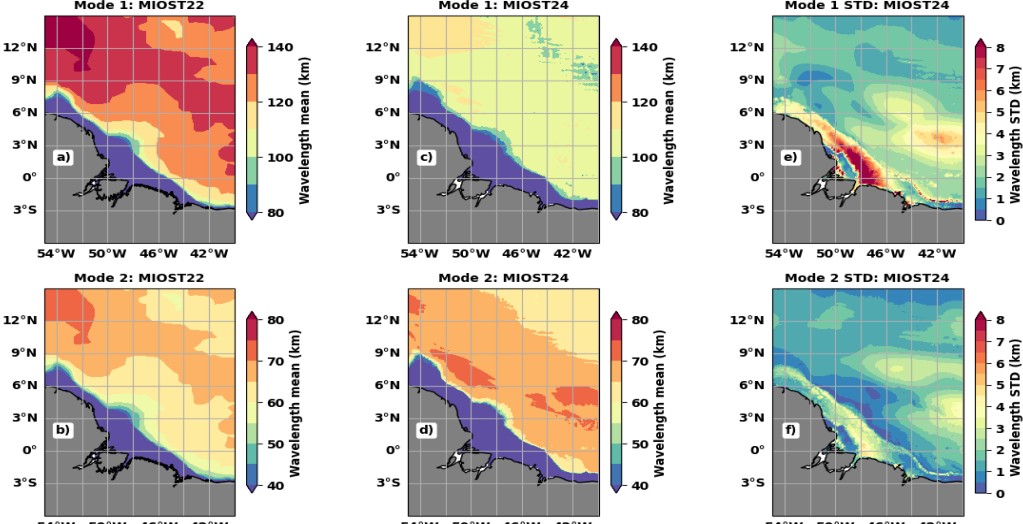


**Figure 3:** As Figure 2 for the region off the Amazon shelf.
The differences between the wavelengths of MIOST22 and MIOST24 reflect the differences between the
stratifications and therefore between the climatological eigenspeed used in each case. According to Equation
1, a velocity difference of 0.1 m/s corresponds to 4 to 5 km shift in the wavelengths for the latitudes of
interest here, between 10°S and 20°N. We have found differences of up to 0.5 m/s between the eigenspeed
of MIOST22 and MIOST24 (not shown). Despite the differences with the MIOST22 wavelengths, we remain
confident about the MIOST24 wavelengths as they are close to those used by Zhao (2018), which is based on
the same method but on the stratification from the World Ocean Atlas 2013 climatology.
The standard deviations of the monthly M2 wavelengths used to derive the MIOST24m atlases are shown
for modes 1 and 2 in the last column of Figures 2 and 3. In both regions, the M2 wavelength variation over
the year is up to 5 km for mode 1 (Figures 2e and 3e) and 3 km (Figures 2f and 3f) for mode 2 in the deep
ocean. On the continental shelves the M2 wavelength variation is up to 8 km for mode 1 and 5 km for mode
2. In the Luzon Strait (Figure 2), wavelength variations are more pronounced in the western part, which opens
to the South China Sea, than in the eastern part, which opens to the Philippine Sea. Wavelengths are relatively
stable in the Celebes Sea (Figure 2), with monthly variations of less than 2 km for mode 1 and less than 1 km
for mode 2. Outside the continental shelf, wavelengths vary more in the eastern part of the Amazon region
(Figure 3). The wavelength variability would also have been biased if the mode 2 wavelengths had been taken



to be half the mode 1 wavelength as in MIOST22. For example, south of the Philippine Sea, mode 1
wavelengths vary between 1 and 4 km (Figure 2e), while mode 2 (Figure 2f) is relatively stable (less than 1
km). In addition, for the boxes in Figure 1a and the region off the Amazon shelf, the spatial means of monthly
wavelengths have been plotted and presented in the Appendix to illustrate further the annual cycle of mode
1 and 2 wavelengths.
**1.3- Database organization for annual and monthly atlases:**
In this study, we use Level 3 along-track data with a resolution of 1 Hz (about 7 km), processed according
to the DT2024 protocol (Kocha et al., 2023) and available on Copernicus website (see link in Data availability).
The HRET correction is reintroduced to obtain an SLA with the full internal tide signal. As in MIOST22, data
are from the Topex/Poseidon, Jason-1, Jason-2, Jason-3, Sentinel-3A, Sentinel-3B, Saral/AltiKa, Cryosat-2,
ERS-1, ERS-2, Envisat, Geosat Follow-On and HY-2A altimetry missions. The altimetry SLAs are selected from
1993 to 2023 and divided into two periods. The MIOST24a and MIOST24m atlases are derived from the SLA
over "period 1" between January 1993 and December 2017, as are MIOST22 and HRET. The "period 2" from
January 2018 to December 2023 is used for validation as independent measurements. The validation consists
of an inter-comparison of the levels of altimetry residual SLA after applying the internal tide corrections using
either the HRET, MIOST22 or MIOST24 atlases.
The 25 years of SLA data from period 1 are used as a single set to derive the annual MIOST24a atlas
(stationary part over 25 years). Period 1 and 2 SLA data are divided into monthly subsets to derive the
MIOST24m atlas and to validate the four atlases. Due to tidal aliasing in the altimetry observations and the
low repeatability of the satellites, the monthly subsets are formed by overlapping the month in question by
15 days on each side. For example, the SLA subset data for April is defined with SLAs from 16 March to 15
May. Finally, the monthly time series covers about 4 years (1500 days) of observations (60 days per year over
25 years), which is a minimum for separating the S2 and M2 harmonics for the Topex/Poseidon and Jasons
missions, whose reference orbit has a period of 9.92 days. After analysis of the altimetry data, the MIOST24a
and MIOST24m M2 atlases are produced on 1/20° horizontal grids, the same resolution as HRET, while
MIOST22 was produced on a 1/10° grid. The HRET mask is applied to the MIOST atlas to harmonize the
comparisons.
**2- MIOST24a M2 atlas, comparison with MIOST22 and HRET:**

This section compares the annual M2 atlas MIOST24a with the annual M2 atlases MIOST22a and HRET.
First, the internal tide amplitude maps M2 are compared. Then, considering the amplitudes $(A, A_m)$, and
phases $(\varphi, \varphi_m)$, the Root Mean Square Error (RMSE) between MIOST24 and MIOST22 or MIOST24 and HRET
is calculated according to equation 3 for a more complete quantification of the distance between the atlases.
Equation 3 can be rewritten as equation 4, where the terms on the right indicate the contribution of the
amplitude differences to the RMSE $(Ac)$ and the contribution of the phase differences to the RMSE $(Pc)$. The
phase differences contribute more to the RMSE when the ratio of the second term to the first term on the
right is greater than 1 $(Pc / Ac > 1$, in equation 5). This ratio is referred to as the rate of the contributions
to the RMSE $(Rc)$ and is evaluated for locations with amplitude differences greater than 3 mm, more precisely
$(A - A_m)^2 > 0.1$, and this is to avoid noise effects on the ratio.
$$RMSE = \sqrt{\frac{1}{2}|Ae^{i\varphi} - A_m e^{i\varphi_m}|^2}, \ (3)$$

$$2RMSE^2 = (A - A_m)^2 + 2AA_m (1 - cos (\varphi - \varphi_m)), (4)$$

$$Rc = Pc / Ac = 2AA_m (1 - cos (\varphi - \varphi_m)) / (A - A_m)^2, (5)$$



The amplitudes of the total internal tides (including mode 1 and 2) and mode 2 internal tides, as obtained for MIOST24a, are shown in Figures 4a and 4b, respectively, for the Indo-Philippine archipelago. Three main areas stand out for their maximum amplitude of more than 8 cm for the total internal tides and more than 4 cm for mode 2: the Luzon Strait, the Sulu Island chain and the Serem Sea. These are the main sources of internal tides in this part of the ocean. Once generated, the internal tides propagate through the various seas of the archipelago. From the Luzon Strait, the intrusion of the internal tides is stronger to the east in the Philippine Sea compared to the west in the South China Sea (Figure 4a).

MIOST24a (Figure 4a) agrees with MIOST22a (Figure 4c) and HRET (Figure 4d) in the spatial distribution of the M2 internal tide amplitude. Compared to the latter two atlases, MIOST24a shows stronger amplitudes and well-defined small-scale structures, because mode 2 is better defined in MIOST24a. On the RMSE maps, the differences between MIOST24a and MIOST22a are 1-2 cm south of 5°N and around Luzon Strait (Figure 4e). For the same locations, the RMSE between MIOST24a and HRET (Figure 4f) increases to 3 cm and non-zero RMSE marks are more visible in the Philippine Sea. The spatial mean of $Rc$ is 1.36 for the RMSE between MIOST24a and MIOST22a, indicating that the atlases differ mainly by their phases. For HRET, the spatial mean of $Rc$ is 1.06, thus amplitudes and phases contribute equally to the differences between MIOST24a and HRET.

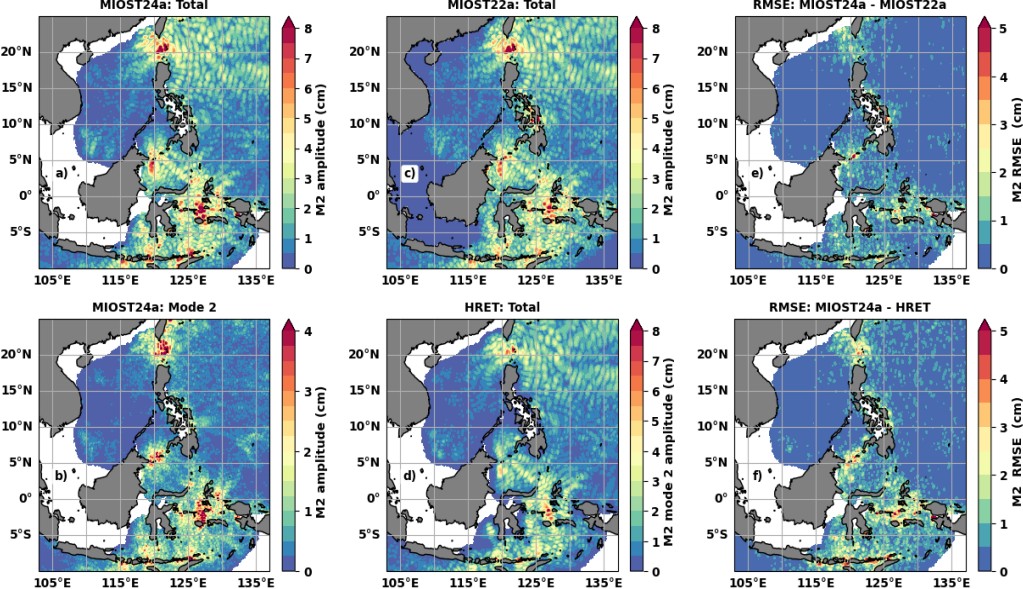

**Figure 4:** Amplitude (a-d, in cm) and RMSE (e-f, in cm) of the M2 internal tides in the Indo-Philippine Archipelago. Amplitudes are shown for (a) MIOST24a total internal tides, (b) MIOST24a mode 2 internal tides, (c) MIOST22a total internal tides and (d) HRET total internal tides. The RMSE is computed for (e) MIOST22a and (f) HRET against MIOST24a. White corresponds to the HRET mask applied to MIOST24 to facilitate comparison.

In the region off the Amazon shelf (Figure 5), MIOST24a is also characterized by offshore propagation of internal tides. Two lines of maximum internal tides are distinguishable south of 2.5°N (Figure 5a), and the third is less clear. Mode 2 from the Amazon shelf-break can be seen following these lines (Figure 5b). Again, the amplitude of the M2 internal tides is stronger in MIOST24a, but the atlas matches MIOST22a (Figure 5c) and HRET (Figure 5d), even if the latter shows smoother structures. As in the Indo-Philippine archipelago, the RMSEs between MIOST24a and the other two atlases are mostly lower than 2 cm but exceed 3 cm at a few locations (Figures 5e and 5f). The MIOST24a atlas differs more from MIOST22a (Figure 5e) and HRET (Figure



5f) in the first 100 to 200 km from the shelf break. The RMSEs are characterized by small scale structures in
this zone where the mode 2 amplitude is maximum. This suggests that the RMSEs reflect the changes
associated with improved mode 2 in MIOST24. The spatial means of $Rc$ are 1.35 and 0.69 for the RMSE
between MIOST24a and MIOST22a and HRET, respectively. In this region, the phase differences are dominant
between MIOST22a and MIOST24a, while the amplitude differences are dominant between MIOST24a and
HRET.

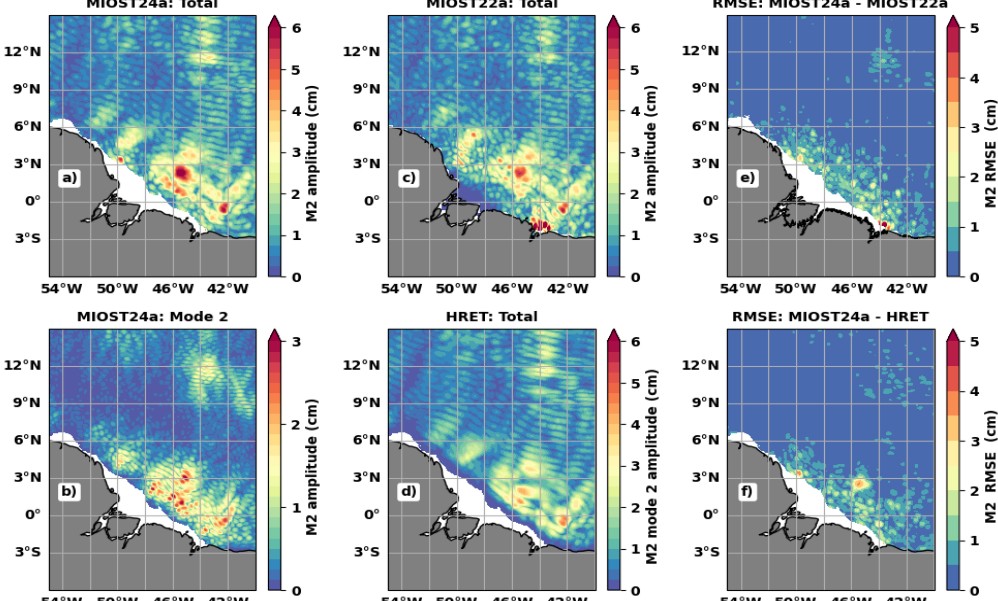

**Figure 5:** As Figure 4 for the region off the Amazon shelf.
**3-  MIOST24m M2 atlases, the monthly variability of the internal tide:**

As a reminder, the MIOST24m monthly atlases are derived from the monthly climatology of M2
wavelength (Section 1.2) and monthly subsets of the altimetry data (Section 1.3). Monthly mode 1 and 2
atlases for April, July, September and December have been selected to illustrate the different propagation
situations and changing amplitudes of internal tides that occur throughout the year. The RMSE is calculated
between the monthly atlases MIOST24m and the annual atlas MIOST24a. The annual cycles of the spatial
mean of the RMSE and $Rc$ are used to investigate the monthly variability of the M2 internal tide.
In the Indo-Philippine archipelago (Figure 6), the propagation of the mode 1 internal tides towards the
Philippine Sea is most pronounced in the early months of the year, such as April (Figure 6a) and in December
(Figure 6d). For the rest of the year, the structure of the internal tides is distorted quite rapidly, as in June
(Figure 6b) and September (Figure 6c). In the South China Sea and the rest of the archipelago, the amplitude
and trajectories of the internal tides vary. Mode 2 variability is more characterized by amplitude variations
(Figure 6 e-h). In the first half of the year in the region off the Amazon shelf (Figure 7), the internal tides of
mode 1 propagate freely from the Amazon shelf-break towards the open ocean, as in April (Figure 7a) and
June (Figure 7b). In the second half of the year, mode 1 has difficulty crossing 5°N, as in September (Figure
7c) and December (7d). The amplitude variations of mode 2 (Figure 7 e-h) are also easily distinguishable in
this region.



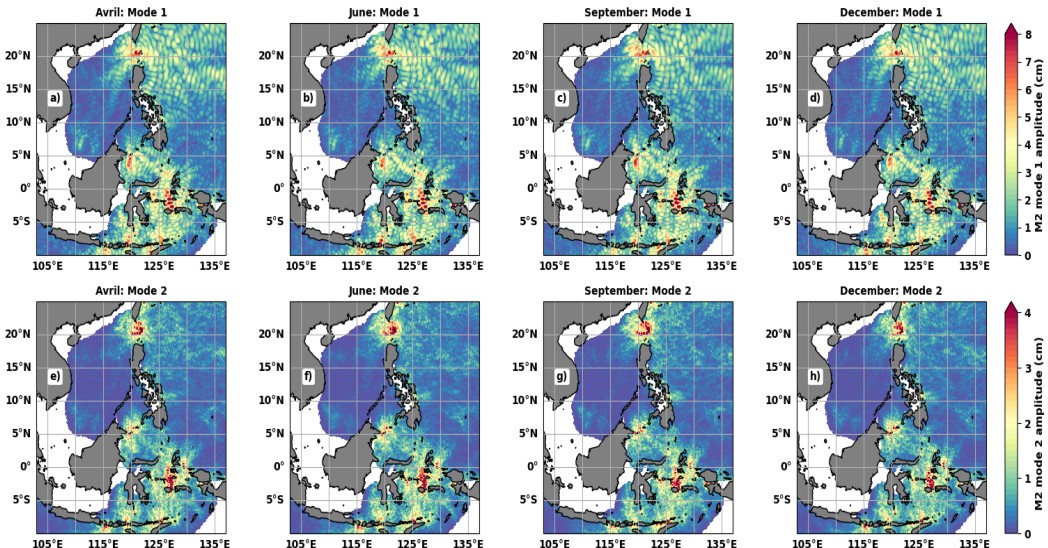

**Figure 6:** MIOST24m M2 mode 1 (top) and mode 2 (bottom) amplitudes (in cm) for Avril (a and e), June (b and f), September (c and g) and December (d and h).

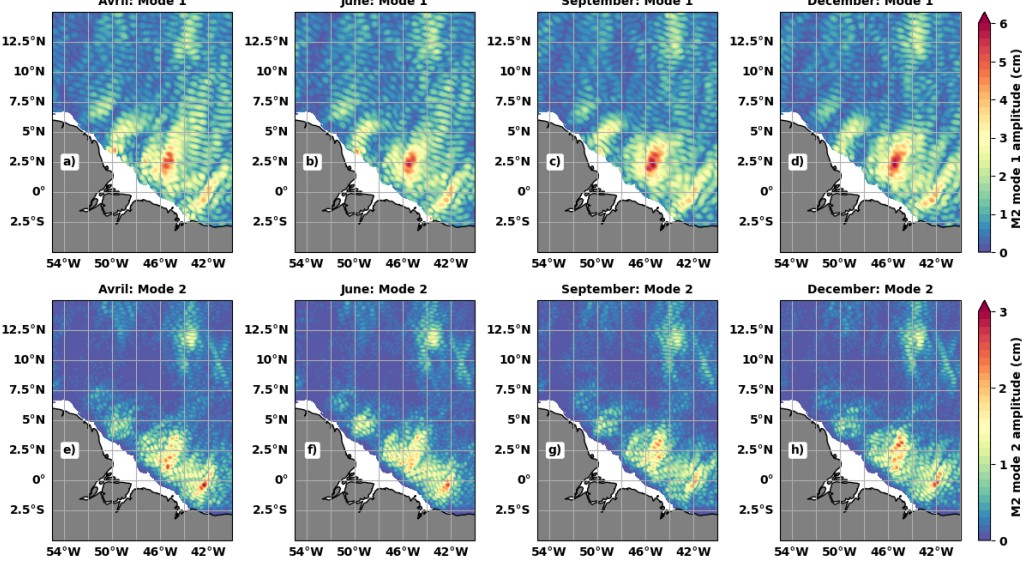

**Figure 7:** As Figure 6 for the region off the Amazon shelf.

The annual cycles of the spatial mean of monthly RMSE (Figure 8) of the total internal tides (in red), mode 1 (in blue) and mode 2 internal tides (in green) confirm that the internal tides are not constant throughout the year. For the two regions of interest, the annual cycle of the spatial mean of the RMSE of the total internal tides is like that of the mode 1 internal tides. The largest discrepancies between the annual MIOST24a atlas and the monthly MIOST24m atlases of the total and mode 1 internal tides occur in January, August and December in the Indo-Philippine archipelago (Figure 8a). Monthly and annual atlases are closest in May





(Figure 8 a). The annual cycle of the spatial mean of the RMSE for Mode 2 is consistent with that of mode 1,
although there is a one-month lag between the maximum peaks of mode 1 and mode 2 (Figure 8a).
Off the Amazonian shelf (Figure 8b), the annual cycle of the spatial mean of the RMSE of the total and mode
1 internal tide is bimodal, with the minimum of July dividing the year into two parts. For mode 2, the RMSE
is relatively constant between January and April, so on average mode 2 internal tides barely vary during these
months. From April to December, the RMSE cycle is bimodal as for mode 1, with peaks in June and September.
The bimodal shape of the annual cycle of the spatial mean of the RMSE is consistent with the behavior of the
M2 amplitudes during the year, as partially shown in Figure 7. It is also reminiscent of the distinction made
by Tchilibou et al, 2022, which separates the MAMJJ (March to July) months of high internal tides coherence
from the ASOND (August to December) months of high internal tides incoherence due to the increase of the
eddy kinetic energy.

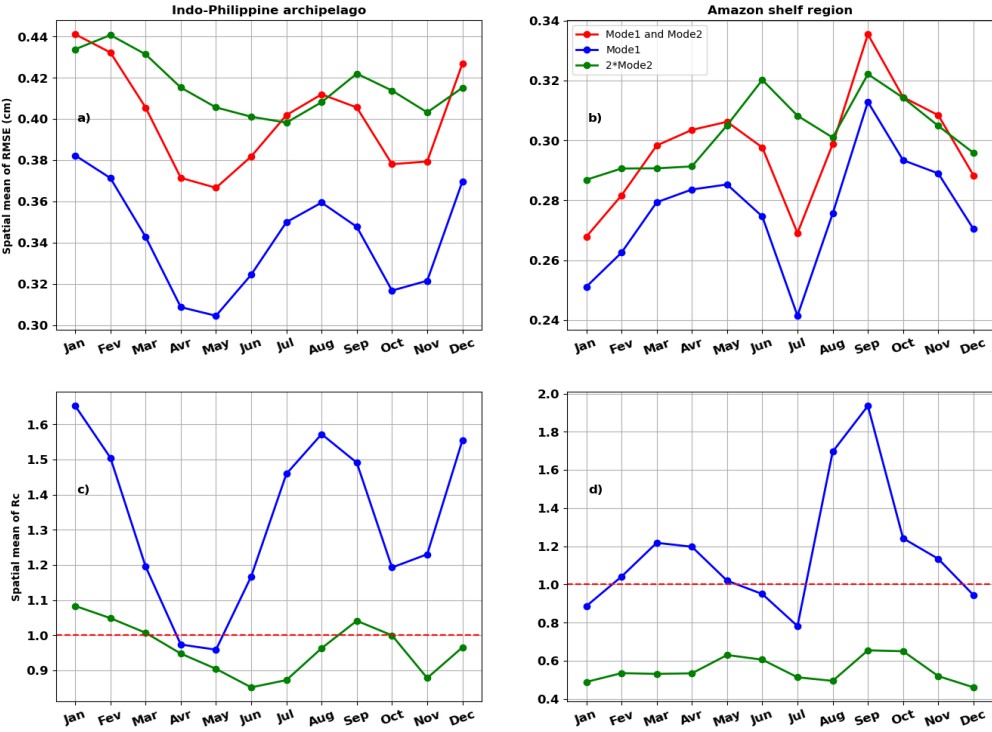


**Figure 8:** Annual cycle of the spatial mean of (top) the monthly RMSE (in cm) and (bottom) the monthly rate
of the contributions to the RMSE ($Rc$) in the Indo-Philippine Archipelago (a and c) and the region off the
Amazon shelf (b and d). The RMSE is calculated for the total internal tide (modes 1 and 2, in red), mode 1 (in
blue) and mode 2 (in green). Note that the RMSE for mode 2 has been multiplied by 2 to improve visibility.
The rate of the contributions to the RMSE ($Rc$) is evaluated following equation 5 and for locations with
amplitude differences greater than 3 mm.

The monthly spatial mean of $Rc$ for mode 1 is greater than 1 for most of the year, except for April and May
in the Indo-Philippine Archipelago (Figure 8c) and especially for July in the region off the Amazon shelf (Figure
8d). The latter months are those with the lowest RMSE (Figure 8a-b). Thus, the RMSEs between the monthly
MIOST24m and annual MIOST24a mode 1 atlases express differences in amplitude for the month with the
minimum RMSE, but in general over the year, differences in phase and therefore in the internal tides
structures between the atlases. For Mode 2, $Rc$ indicates that the RMSE reflects that amplitude differences



predominate over phase differences in the two regions. Except, of course, for January, February and
September in the Indo-Philippine archipelago, for which the RMSE is maximum (Figure 8a).
Examples of the annual cycle of the M2 internal tides amplitude are shown in the Appendix. These cycles
are not identical to those of the RMSE, as the approach is different. However, they confirm that using
altimetry data and MIOST provides access to the monthly variability of the internal tide. In the next section,
we will test the effectiveness of monthly atlases in removing the internal tides from altimetry observations.
**4-  Impact of internal tide correction in altimetric data: comparison between monthly and annual**
**atlases.**
The variance reduction is a way of measuring how much an internal tide atlas removes or corrects the
internal tides in the altimetric SLA. In equation 6, the variance reduction is defined as the difference between
the variance of the SLA corrected with the M2 internal tide prediction and the variance of the uncorrected
SLA. In this way, a negative value of the variance reduction means that the internal tide atlas has helped to
reduce the variance of the SLA. On the other hand, a positive value of the variance reduction means that the
internal tide atlas had the opposite effect to what was expected and increased the variance of the SLA.
$$variance\ reduction = var\ (SLA - M2\ prediction\ ) - var\ (SLA), (6)$$
The annual M2 HRET, MIOST22a, MIOST24a and the monthly M2 MIOST24m atlases are each used in turn
as the M2 correction in equation 6, and the monthly variance reductions are calculated in pixels of 0.5° x
0.5°. As the main objective is to test the robustness of the monthly atlases, the variance reductions are
estimated each month on monthly subsets (see section 1.2) of the altimetry data from period 1 (1993-2017)
and period 2 (2018-2023). Examples of variance reduction maps are shown for September only. The atlases
are then classified according to the annual cycles of the spatial mean of the monthly variance reduction.

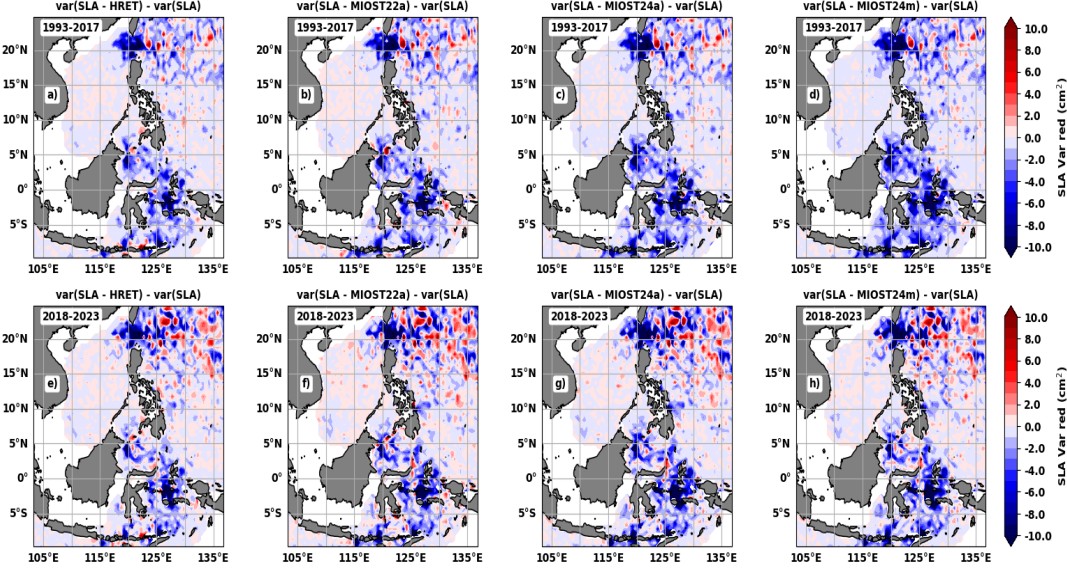


**Figure 9:** SLA variance reduction (in cm2) in the Indo-Philippine archipelago during September for period 1
(a-d, 1993-2017) and period 2 (e-h, 2018-2023).





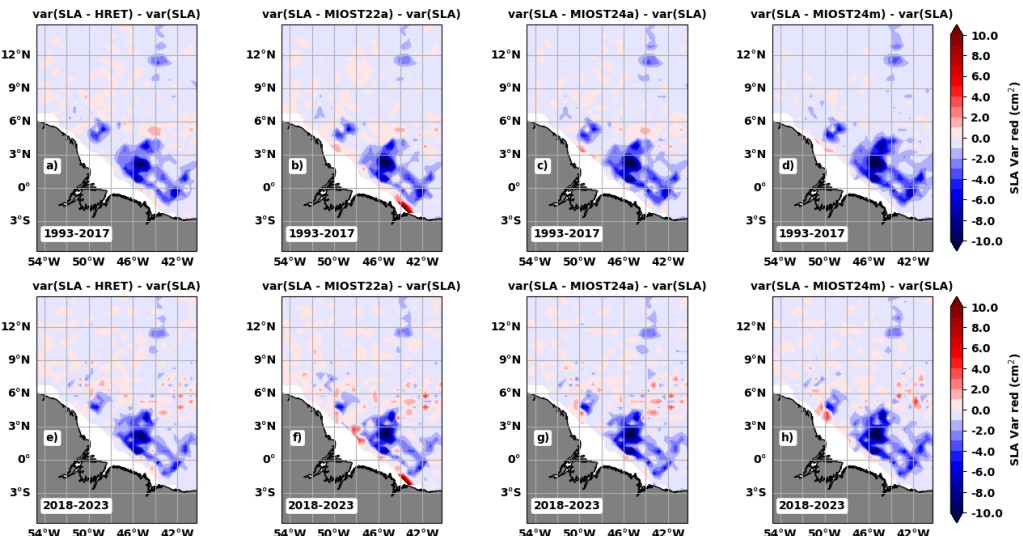


**Figure 10:** As Figure 9 for the region off the Amazon shelf.
In September (Figure 9), as in other months of the year (not shown), the four atlases performed particularly
well around the internal tide generation zones in the Indo-Philippine archipelago, including the Luzon Strait
and the entire area south of 10°N (Figure 6).In the Amazon region (Figure 10), the beneficial effects of the
atlases in reducing the SLA variance are noticeable in the areas and beams of maximum amplitude of the
internal tides previously shown in Figure 7.  Some areas, such as the Philippine Sea (Figure 9) and the Amazon
shelf-break (Figure 10 b and f), present a local increase in variance due to the internal tides atlases. Compared
to period 1 (Figure 9 a-d and Figure 10 a-d), this negative effect of the atlases is more pronounced in period
2 (Figure 9 e-h and Figure 10 e-h). However, in period 2 the atlases still perform well at the locations where
the internal tides are generated and along their main trajectories. This indicates that in period 2 the atlases
have more difficulty in correcting the incoherent internal tide.
In both regions, the MIOST24m atlas has the best internal tide correction in period 1 (Figure 11 a-b),
followed by the MIOST24a atlas. The atlases that eliminate the least variance in period 1 are HRET for the
Indo-Philippine archipelago (Figure 11a) and MIOST22a (Figure 11b) for the region off the Amazon shelf. The
positions of MIOST24a and MIOST24m are reversed in period 2 in the Indo-Philippine archipelago (Figure
11c), with MIOST24a being slightly better for most months. In the Amazon region (Figure 11d), the MIOST24m
atlas outperforms the other atlases between March and June and again between August and November. The
positions of MIOST22a and HRET in period 1 are unchanged in period 2.
It is not surprising that the MIOST24 atlases performed better than HRET and MIOST22a in period 1, the
variance reductions are indeed calculated on the same dataset used to obtain the MIOST24 atlases. However,
the fact that the two MIOST24 atlases perform better on independent period 2 data shows that the
wavelengths' change from MIOST22 to MIOST24 has a positive impact on the internal tide correction in the
altimetry data. Regarding the use of monthly atlases to correct internal tides in altimetric SLA, our results
show that they are appropriate and indicated when applied to the data used to construct them. On an
independent dataset, the monthly atlas remains effective at the generation site and on the main trajectories
of the internal tide, but it may struggle to correct the internal tides on the secondary trajectories associated
with the incoherence of the internal tide. Finally, MIOST22a is the least efficient atlas in the Amazon region
because it increases the variance of the SLA along the coast, and in the open ocean it is better than HRET but
not than the MIOST24 atlases.



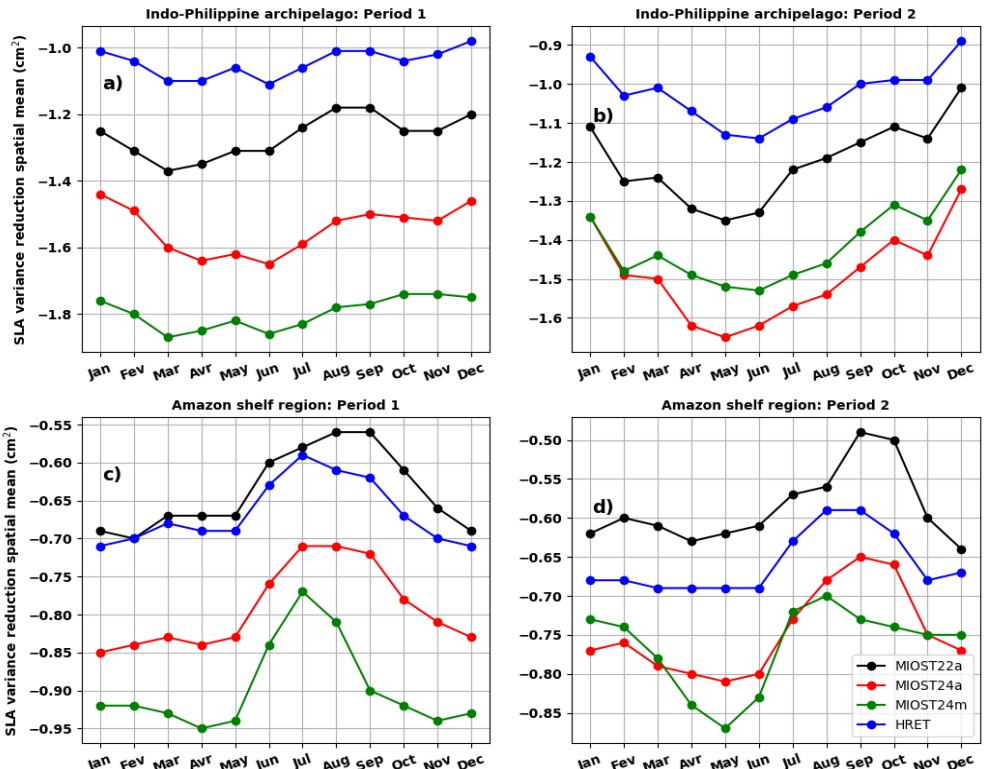


**Figure 11:** Spatial mean of monthly variance reduction (in cm²) in (left) the Indo-Philippine Archipelago and (right) the region off the Amazon shelf in (top) period 1 (1993-2017) and (bottom) period 2 (2018-2023). The variance reduction was calculated according to equation 6, using either M2 prediction from the MIOST22a (black), HRET (blue), MIOST22a (red) and MIOST24m (green) atlases.

### 5- Conclusion.

This study presents the M2 MIOST24 (MIOST 2024) internal tide atlas for modes 1 and 2 in the Indo-Philippine archipelago and the region off the Amazon shelf, derived from 25 years (1993-2017) of altimetric SLA and compared with the existing atlases MIOST22 (version 2022, Ubelmann et al., 2022) and HRET (Zaron, 2019). The latest is currently used as the reference atlas for correcting internal tides in altimetry data. The improvement of M2 wavelengths is the distinguishing feature between MIOST24 and MIOST22. MIOST24 uses the M2 wavelengths of modes 1 and 2 obtained after solving the vertical mode equations constrained by the stratification of the 1993-2020 monthly climatologies from the GLORYS12v1 reanalyzes, whereas MIOST22 used the first Rossby deformation radius climatology of Chelton et al. (1998) as an approximation of the mode 1 wavelength, then divided the mode 1 wavelength by 2 to obtain the mode 2 wavelengths. The MIOST24a annual atlas is produced by averaging the GLORYS12v1 monthly reanalyzes and using the altimetry data as a set. In contrast, the MIOST24m monthly atlases are derived from GLORYS12v1 monthly reanalyszes and subsets of altimetry data, defined with a 15-day overlap in each of the months adjacent (see section 1.3) to the month of interest. The differences between MIOST24, MIOST22 and HRET, the monthly variability of the internal tide and the relevance of using a monthly atlas to correct for the internal tide in altimetry data are discussed.



The RMSE calculated between M2 atlases of MIOST24a and MIOST22a and between MIOST24a and HRET
reaches 3 cm at certain locations in the Indo-Philippine archipelago and the Amazon region, highlighting
internal tide estimates that were certainly biased in previous atlases. The spatial mean of the rate of
contributions to the RMSE ($Rc$), defined as the ratio between the contribution of phase differences to the
RMSE and the contribution of amplitude differences to the RMSE, shows that the differences between
MIOST24a and MIOST22a are mostly explained in the two regions of interest by the phase differences
between the two atlases. This means that the differences between the internal tide patterns of MIOST24a
and MIOST22a are greater than the differences between the amplitudes of these atlases. This is not surprising
as the M2 internal tide structures are better represented in MIOST24a thanks to the improved wavelengths
of modes 1 and 2 in MIOST24. The differences between HRET and MIOST24a are related to both amplitude
and phase differences. This is because the internal tide amplitudes are larger in MIOST24a and HRET is a
smoothed atlas.
The amplitude maps of the MIOST24m atlas and the monthly RMSE calculated between MIOST24a and
MIOST24m confirm that the M2 internal tides vary over the year. In the Indo-Philippine archipelago, mode
1 internal tides generated in the Luzon strait propagate strongly towards the Philippine Sea in the early
months of the year (such as April) and in December. In other months, such as June and September, the
internal tide structure deforms rapidly. In the South China Sea and the rest of the archipelago, the amplitude
and trajectories of the internal tides show significant monthly variability. The total (including modes 1 and 2)
internal tide atlases for January and August are the least close to the annual atlas, whereas May is the closest.
Mode 2 atlases are distinguished by their amplitude, being furthest from the annual atlas in February and
September and closest in July . Overall, for this region, the variability of the monthly atlases has yet to be
linked to changes in stratification, seasonal cycles of currents (Sprintall et al., 2019) and seasonal cycles of
cyclonic and anticyclonic eddies (Hao et al., 2021), this is beyond the scope of this study and needs special
future dedicated study to be revealed.
In the Amazon region, our results on the monthly variability of M2 internal tides are in good agreement with
those of Tchilibou et al. (2022): from March to June the internal tides propagate far offshore, whereas from
August to December the propagation seems to stop at about 5°N. Tchilibou et al (2022) have shown that this
is the result of the interaction between internal tides and the mesoscale, which is stronger in the fall season.
The RMSE between MIOST24a and MIOST24m allows us to find the bimodal variation MAMJJ (March to July)
and ASOND (August to December) of Tchilibou et al. (2022) for mode 1 and the total internal tide, and to
show that mode 2 remains relatively stable until April before following the bimodal cycle of mode 1.
The annual cycle of the spatial mean of $Rc$ suggests that, in the Indo-Philippine archipelago and the region
off the Amazon shelf, for mode 1, the main differences between the annual and monthly atlases result from
changes in the phase (*e.i.* the spatial distribution of the M2 internal tides). For mode 2, the annual and
monthly atlases are characterized by differences in the internal tide amplitude. The results from these two
regions illustrate that altimetry can be used, to some extent at least, to study the monthly variability of the
internal tides. In a few years, the monthly atlases could be re-evaluated with longer data series and limiting
the overlap of the data from the surrounding months, which may improve furthermore the present results.
The last part of the study is devoted to the quantification of the new internal tide correction in the altimetry
data : to date, annual atlases (HRET, MIOST22a or MIOST24a) are used instead of monthly atlases
(MIOST24m), which are more accurate in terms of the internal tide variability. Variance reductions were
calculated for the period (1993-2017, period 1) used to derive the MIOST24 atlases and for an independent
period (2018-2023, period 2). For both regions, MIOST24m is the atlas that best corrects the internal tide
signal in period 1 altimetry data. In period 2, MIOST24a is better in the Indo-Philippine archipelago, while
MIOST24m is better in the region off the Amazon shelf depending on the season. As MIOST24m is not always
the best atlas for period 2 in both regions, it is more judicious to choose between annual and monthly atlases
depending on the objectives. Monthly atlases are recommended if they are to be applied to the same



database from which they were derived as they reflect a specific seasonal variability which can have strong
annual variations. On independent data, monthly atlases should work well around the internal tides
generation sites and along their main trajectories. Beyond these locations, the performance of the monthly
atlases is conditioned by changes in the secondary trajectories reflecting the incoherence of the internal tide.
The annual atlas, on the other hand, may underestimate the correction around the generation sites, but
make a less degraded correction along the secondary trajectories of the internal tide.
In general, the new MIOST24 M2 atlases extract internal tides from altimetric data better than MIOST22
and HRET for the Indo-Philippine and Amazon regions. The results of MIOST24 are encouraging and justify
the development of a global version of the annual MIOST24 atlas for the M2 wave as well as for the N2, S2,
K1 and O1 waves. The new MIOST24 atlas could incorporate satellite data beyond 2017. Initially, the new
global atlas will only include the classic nadir missions described in this article. In a second phase, tests could
be carried out to incorporate 2D observations from the new SWOT (Surface Water and Ocean Topography)
KaRIn mission to improve the spatial resolution of the atlases and its impact on the mode 2 descriptions.

**Appendix:**
**A.1- Example of an annual cycle of mode 1 and 2 M2 wavelengths**
The curves of the spatial averages of the monthly M2 wavelengths of mode 1 (in black) and mode 2 (in
green) are shown in Figure A1. The spatial means are calculated in the Luzon Strait (Figure A1a), Sulu Island
Chain (Figure A1b) and Seram Sea (Figure A1c) boxes as shown in Figure 1. The Amazon region is treated as
a box because internal tides generation occurs over almost the entire shelf break, although it is dominant in
the six locations shown in Figure 1. The annual cycles of mode 1 and 2 M2 wavelengths are in phase in the
Luzon Strait (Figure A1a). Mode 1 increases by an average of 7 km between February (when it is lowest) and
September (when it is highest). For mode 2, the variation is about 5 km. In the Sulu Island chain (Figure A1b)
and for mode 1, a first peak is observed in June and a second peak in October. Although the cycle of mode 2
is close to that of mode 1, the monthly variations remain small (less than 1 km, Figure A1b green curve) and
agree with Figure 2f.

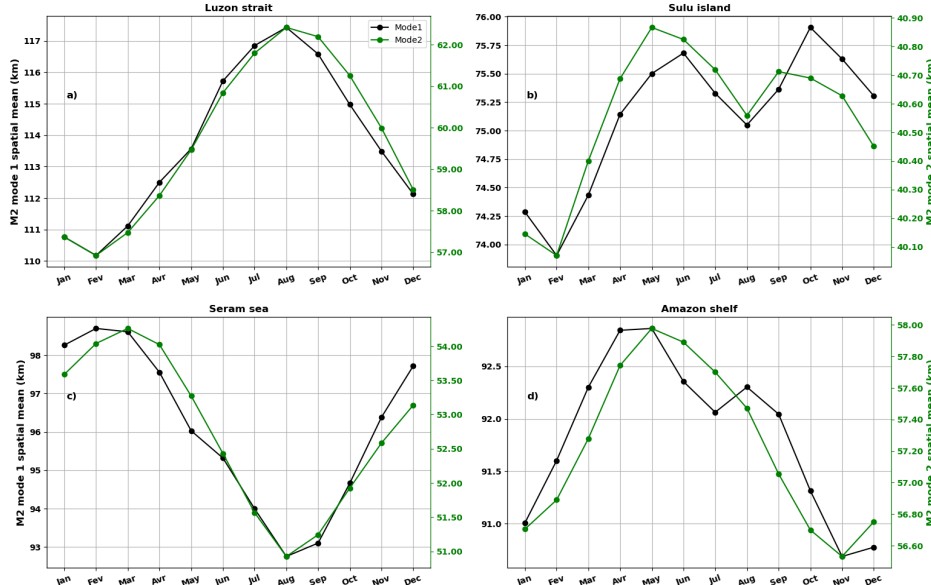




**Figure A1:** Annual cycle of mode 1 (black) and mode 2 (green) M2 internal tide wavelengths in the Luzon Strait (a), the Sulu Island chain (b), the Serem Sea (c) and the Amazon region (d). The right vertical axis (green) corresponds to mode 2.

The annual cycle of M2 wavelengths in the Seram Sea (Figure A1c) is opposite to that observed in the Luzon Strait. The longest wavelengths are observed in March and the shortest in August, representing a variation of more than 4 km for mode 1 and about 3 km for mode 2 (Figure A1c). In the Amazon region (Figure A1d), there are variations of about 2 km for mode 1 and 2 wavelengths, which are the longest between April and May and the shortest in November. However, the spatial mean is certainly influenced by variations in the eastern part of the basin and reduced by small variations in the western part of the basin (Figures 3e and 3f).

### A.2- Annual cycle of M2 internal tide amplitude

The annual cycles of the internal tide amplitude M2 are obtained by averaging, as above, the amplitudes of the monthly MIOST24m atlases in the Luzon Strait, the Sulu Island Chain and the Serem Sea for the Indo-Philippine archipelago, and the entire Amazon region. The annual cycles of M2 internal tide amplitudes are represented by black, dashed black and green curves for total tide (mode 1 and 2), mode 1 and 2, respectively.

In the Luzon Strait (Fig. A.2a), the amplitude of the internal tides reaches its maximum in October. This maximum of about 4 cm follows a relative maximum of 3.6 cm that occurs in spring (April-May), giving the annual cycle a bimodal appearance. The amplitude of mode 2 is maximum in August and minimum in February. The apparent bimodality of mode 1 could be related to the combination in our Luzon Box of the annual antiphase cycles of the M2 internal tides propagating eastward in the Pacific (Philippine Sea) and westward in the South China Sea, as described in Zhao and Qui (2023). The intense eastward flow in winter and spring is expected to contribute to the relative maximum, while the intense westward flow in summer and autumn helps to increase the mean amplitude to the absolute maximum.

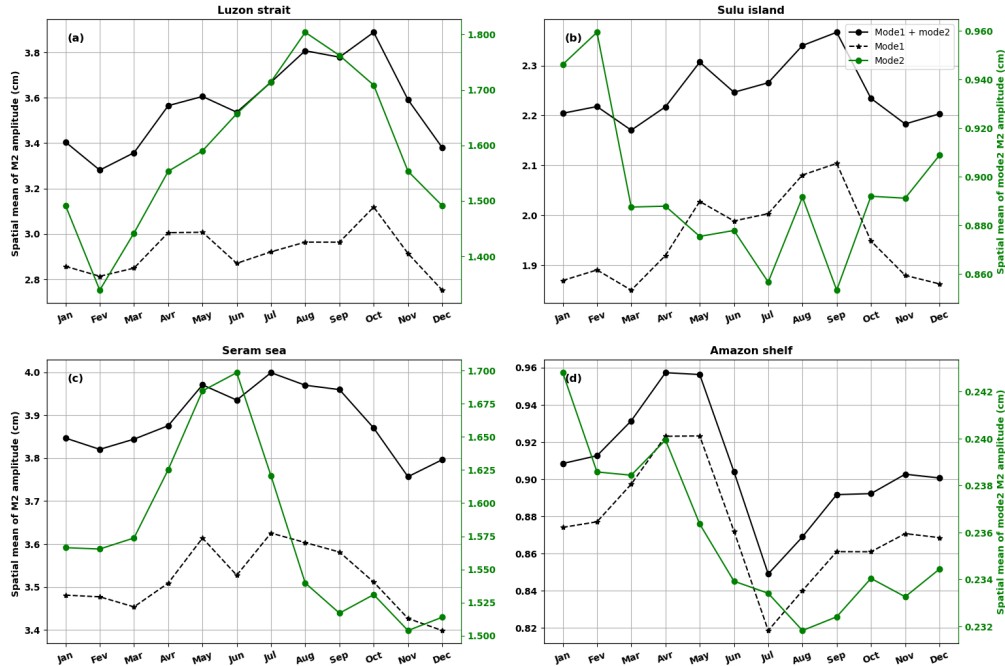



**Figure A2:** Annual cycle of M2 internal tidal amplitudes in the Luzon Strait (a), the Sulu Island Chain (b), the Serem Sea (c) and the Amazon region (d). In black is the total amplitude including modes 1 and 2, dashed black is mode 1, green is mode 2 and the vertical axis on the right (green) corresponds to mode 2.

In the Sulu Island chain, the annual cycle is bimodal for the total internal tides and mode 1 (Figure A2b). The peaks occur in May and September and are at almost identical levels. The amplitude of mode 2 remains relatively constant, as do the wavelengths of this mode. It cannot be excluded that the shape of the annual cycle of mode 1 in this part of the ocean is related to seasonal divergences in the propagation of internal tides towards the Sulu and Celebes Seas, as observed in Figure 6. In the Seram Sea, the amplitudes of the total and mode 1 internal tide reach their maximum in July, one month later than the mode 2 maximum (Figure A2c). In the second half of the year, the amplitude of the internal tides decreases. The annual cycle is closer to monomodal, although the first maximum for mode 1 occurs in May.

In the Amazon region (Fig. A2d), the internal tides (total and mode 1) are highest between April and May and lowest in July. The distribution of the monthly RMSE in Figure 8b and Tchilibou et al. (2022) is somewhat recovered. The M2 amplitudes are very strong between March and June when the internal tides propagate freely. The weak amplitudes between August and December form a second block of variation. The spatial averages of the mode 2 M2 internal tide amplitude vary between 0.235 and 0.245 cm in the Amazon region. This variation is quite negligible and could be explained by possible compensation for what happens in the different internal tide generation sites. For example, in Figure 7, the transition from April to September is characterized by a decrease in mode 2 amplitude around site B and an increase in amplitude around sites A and E.

All the annual cycles presented confirm that MIOST and altimetry provide some access to the annual variability of the internal tide. However, the cycles presented are sensitive to the area in which the spatial averaging was carried out. They cannot be considered strictly as the annual cycle of the internal tides in these regions. This is one of the reasons why we have not discussed these cycles in the main part of the paper and have not attempted to establish the links between these cycles and variations in ocean stratification and circulation. The main objective of our study remains the production of annual and monthly atlases of internal tides. To gain a better understanding of the annual internal tidal cycles, further studies should be carried out, for example using a high-resolution model that would allow a focus on the locations where internal tides are generated.

**Authors contributions:** This work is part of DUACS-RD and marée-SWOT/SALP projects funded by the CNES at CLS. MT's work and analyses are supervised by LC. Conceptualization: MT, LC, SB. MT wrote the paper with contributions from all co-authors.

**Data availability**: Level 3 along-track altimetric data are available on Copernicus website : https://data.marine.copernicus.eu/product/SEALEVEL_GLO_PHY_L3_MY_008_062/description. The GLORYS12v1 reanalysis is available at https://doi.org/10.48670/moi-00021.

**Competing interests:** The contact author has declared that none of the authors has any competing interests

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
