# Peer review of "M2 Monthly and annual mode and mode internal tide atlases from altimetry"

_EGUsphere, 2024_

## Author Response (AR1)

RC1

We would like to thank the reviewers for reviewing our article and recommending it for publication. Each question is answered after the symbols ">>". The lines of the citation refer to the trackchange manuscript.

Comments

- On lines 114-115, the authors state that "…, for MIOST22 the wavelength of mode 1 corresponds to the first Rossby deformation radius…", but I have doubts about this claim. This is because, in constructing the MIOST22 dataset, Ubelmann et al. (2022) estimated the phase velocity (referred to as "Cp" in equation 12 of Ubelmann et al. (2022)) using a method similar to the one in this manuscript. Equation (1) of this manuscript and equation (12) of Ubelmann et al. (2022) are apparently identical when considering $C_p = \omega \lambda_n / 2\pi$. Also, for the first baroclinic-mode phase speed (referred to as "Cn" in this manuscript and as "c" in Ubelmann et al. (2022)), Ubelmann et al. (2022) used Chelton et al.'s (1998) dataset, which is estimated through the WKB method (equation 2.2 of Chelton et al. (1998)). The WKB method provides a solution close to that of the eigenvalue problem method used in this manuscript (Chelton et al., 1998). Considering the above, the statement that "for MIOST22 the wavelength of mode 1 corresponds to the first Rossby deformation radius" appears to be incorrect. This is a crucial point in the motivation of this study, so I would appreciate an accurate response.

  >> Thank you for pointing out this sentence in the manuscript. It is true that the method used in this manuscript is similar to that used in Chelton et al. (1998), as we both solve a Strum-Liouville problem, but with a different method. The confusion in this sentence stems from the different objectives of the studies (eddy Rossby radius versus internal tide wavelengths), but the baroclinic modal decomposition approach is still valid in both cases. What really changes between the two studies is the stratification used: Chelton et al. (1998) used the World Ocean Atlas 1994 climatology of measured profiles and we used a climatology based on the GLORYS12v1 reanalysis. We have rephrased the sentence to indicate that the phase velocities in Ubelmann et al. (2022) are from the Chelton et al. (1998) dataset. See: "To calculate the mode 1 wavelength, Ubelmann et al. (2022) used the phase speed dataset of Chelton et al. (1998), estimated by the Wentzel-Kramers-Brillouin (WKB) method, which itself approximates the eigenvalue solution." (L54-L56). We also pointed out the stratification used: " Furthermore, the climatology of Chelton et al. (1998) is based on stratification profiles calculated from the temperature and salinity of the World Ocean Atlas 1994 (Levitus, 2013), which refers to a different period from the 1993-2017 period of the altimetry data used by Ubelmann et al. (2022)." (L59-L61)

- The figure caption in Fig. 6 may have a typo: "Avril" should be "April."

  >> Done

RC2

We thank the reviewers for reviewing our article and for recommending it for publication. Each question has been answered after the symbols ">>". The line of the citation refers to the trackchange manuscript.

Detailed comments:

Abstract:

L16: What is MIOST22?

  >> MIOST22 is also an internal tide atlas, so we've added a precision in the text (L18).

L27: What is implied by "improved"? Say, "at monthly resolution", if that is what is meant.

>> The word has been removed; we were talking about developing global versions of MIOST24.

L33: comma or not after "et al."?

>> Comma inserted

L36: comma before Desai. Are these lists of citations formatted correctly? I would recommend a semicolon between citations.

>> The format of the references has been changed. They are now listed in alphabetical order.

L37-38: "Zaron (2019)" not "Zaron et al (2019)"

>> Done

L80: "wavelengths' bases" --> "assumed wavelengths"?

>> Done

L73-L75: here and throughout: I cannot see a pattern in the ordering of multiple citations. Please use a pattern such as chronological, alphabetical, etc.

>> The format of the references has been changed. They are now listed in alphabetical order.

L72-L83: This paragraph should mention the work of Kaur: https://os.copernicus.org/articles/20/1187/2024/L86-L88: fix left margin

>> Thank you for your suggestion, the reference has been added.

L86-L88: fix left margin

>> Done

L117: Is there a reason why the rigid lid condition was used at the upper boundary? I think Wunsch JTech 2013 discusses sensitivity to this assumption.

>> There was no particular reason for our choice. Wunsch (2013) does discuss this assumption, the main difficulty of using it being the impossibility to discriminate barotropic and baroclinic surface elevation. As the SLA signal is already corrected for barotropic tides, we consider the rigid lid condition to be sufficient.

L129: The monthly standard deviations appear to be miniscule, no more than 5% of the mean in the deep water where modes 1 and 2 could conceivably be identified. Is this variability found to be significant?

>> Figures 2 and 3 show the wavelength standard deviations for the whole period 1993 - 2020. They give an indication of the ranges of wavelength variability. On a monthly scale the variability can be more significant (see example in Appendix). Although the percentage may appear small, there are areas of the deep ocean where internal tide variability is effective.

L190: This is very good that you used a long data window for validation, sufficient to distinguish the major tides.

>> Thank You

L193-L203: This paragraph makes it clear that the monthly atlas contains a tidal estimate for each calendar month, averaged ofer the 25 years of the Period 1 dataset. Prior to this paragraph, I misunderstood and thought that the monthly estimates were provided for each and every month in Period 1. This is a difficult distinction to communicate, but I think it is important enough that it should somehow me made earlier in the text. Also, if I am understanding correctly, the monthly harmonic

constants are, in fact, phase-locked or stationary (since they are obtained from the average months over 25 years data). This distinction should also be made.

>> Yes, the monthly harmonic constants are phase-locked. We have added a sentence to the introduction to clarify that this is not the average of the monthly atlases for each year, but an atlas based on 12 subsets of data over 25 years "The 25 years of SLA altimetry data are divided into monthly subsets to produce monthly atlases of coherent internal tide" (L74-L75).

L274: "Avril" --> "April"

>> Done

Fig 6 and Fig 7: These figures are too small and I would like to see more detail. Please use 2-columns max for these.

>> We used 4 columns to avoid making the figures too large. However, they have been slightly extended

Fig 8b: Increase size of legend text.

>> Done

Fig 8 and L286-L294: Instead of using RMSE, maybe you could refer to it as RMSD, since you have not made a definition of what is "error" vs a simple "difference".

>> Thank you for your suggestion. We've replaced RMSE with RMSD throughout the article.

Fig 11: These are nice summary figures. The text also does a good job of distinguishing the performance of the models during Period 1 and Period 2, where it is the latter which is the real test period.

>> Thank you